# OpenReview forum: "Reducing the Scope of Language Models with Circuit Breakers"
_ICLR.cc/2025/Conference — ICLR 2025 Conference Withdrawn Submission_

### Official Review · Reviewer_HDh7 · 2024-10-22

**Soundness:** 2
**Presentation:** 2
**Contribution:** 3
**Rating:** 5
**Confidence:** 4

**Summary:**

The paper tackles the problem of “scoping” language models, that is, restricting their generations to a specific set of tasks while rejecting queries for other tasks. This is a generalization of the well-studied problem of refusal in response to harmful queries. The authors conduct a comprehensive evaluation of the effectiveness of existing methods to scope Mistral 7B, covering different methods (prompting, fine-tuning, DPO, probing, CB) and 9 diverse tasks. The evaluation considers multiple axes – the robustness against adversarial attacks, the diversity of the rejection set, and the number of accepted tasks.

The results show the advantages and limitations of current methods and highlight the robustness of circuit breakers (CB) – a recent method for restricting LLM behavior. Specifically, while CB does not generally outperform other methods, it seems to be the best in terms of generalization to OOD rejection samples, and combining CB with fine-tuning produces a better variant that effectively combines their advantages. The paper also includes interesting analyses in the appendix.

**Strengths:**

* This is a timely and interesting work, which broadens the notion of refusal in LLM into practical scenarios. It is also interesting from the perspectives of interpretability and control.

* The paper is well contextualized with the literature and the related work section is comprehensive.

* Most of the experimental choices are well-justified.

* The finding that CB+SFT complement each other and improve SFT generalization while retaining its performance is nice and opens interesting research directions for future work.

**Weaknesses:**

While I find the paper very interesting, there are multiple concerns regarding the results and presentation.

### The quality of the rejection detectors is not evaluated
The different methods used seem to rely on training heuristic detectors, like keyword-matching or repetitions. How well do these detectors perform in practice? It seems they were trained on very small samples of 30 examples, but how much of the model rejection/acceptance behaviors do they capture?
Specifically, the authors describe cases where the model first rejects a query but then as generation continues it moves to accept it. This behavior is not explicitly measured (even qualitatively). Ideally, for scoping, we would like to see a rejection without moving to acceptance later on. I guess one could simply use a detector and based on that stop the generation, but then no evaluation of that detector was conducted. Therefore, whether a method yields a high rate of such rejection→acceptance cases and how many of those are detected is a factor that should be measured.
Also, the fact that rejection is measured on a heuristic basis and with different detectors for different methods could be noisy. I’d expect to see a qualitative comparison in addition to the quantitative results, or numbers that justify the quality of these detectors on a larger sample.

### Robustness of the results
I am concerned by the fact that the vast majority of experiments are limited to one model and a single system prompt (that was also used in training some of the methods).

* Regarding the model – the vast majority of experiments in the paper are performed on Mistral 7B. Results on an additional model (Granit 7B) for the robustness experiment are provided in the appendix, but seem not to agree with the results for Mistarl. Specifically, CB often does not seem to be robust on OOD samples, which has been highlighted as its main advantage over other methods. Taken together, the observations (at least for this experiment) are not conclusive and show that more evaluation is needed for this experiment.

* Regarding the prompt – This could be problematic since models can behave very differently for paraphrases and perturbations. Examples of the dramatic effect of this on evaluation are https://arxiv.org/abs/2401.00595 and https://arxiv.org/abs/2310.10062.

While I understand the authors’ argument that all tested methods have been evaluated on multiple models, their tradeoffs and performance on scoping were not tested.
If the paper argues for a comprehensive evaluation it should be more careful about these choices. Generally, I think the fact that the results are not conclusive for Mistral and Granit is fine as long as the paper is framed as an evaluation paper. But then it is important to understand those differences, perhaps by extending the robustness evaluation to another model like LLaMA (with different specs, like size and training data).
If the experiments are expensive to run, then it would be valuable to provide at least some evidence that reduces these concerns. For example, an analysis that demonstrates on 1-2 methods that the prompt does not affect the robustness results substantially.


### Readability and presentation

a) Since the paper poses the problem of scoping, I would expect section 4 (results) to start by answering this question, namely, how well do methods perform on scoping? The experiments cover many axes but this question remains unaddressed, until realizing there are some results for this in Appendix A.2 (but it is not referenced at the beginning of section 4).

b) The title makes it sound like this is a paper that proposes a method rather than an extensive evaluation. As such, the reader could expect a more fine-grained analysis of CB performance. I believe the title could be more informative by focusing more on the evaluation part rather than on CB as of now.

c) I found some parts of the paper hard to read, and there are many typos and some ungrammatical or unclear sentences. The main figures are difficult to digest so it is challenging to see the trends and easily understand the results. I now elaborate on all these points.

* It is hard to read Figure 1 before reading section 3, since it is not clear what the reported metrics are.
* The main results in Figures 1 and 2 are hard to parse. It’s difficult to compare the numbers because the bars are small and there is a lot of information, which makes it hard to see the general trends. Consider averaging over the different attacks and report mean and standard deviation results in a table or a simplified plot.
* The beginning of section 4 is redundant to section 3. It is also redundantly repeated many times throughout the papers that the number of experiments is large.
* Example typos:
    * Line 457: “is it possible to still reject tasks well when there are multiple tasks in the rejection set” I believe the authors mean “in the acceptance set”.
    * Line 86: typo “has many broadly performs more robustly”
    * The end of introduction “We show unlike other methods” (missing “that”)
    * Line 164: missing a period at the end of the sentence.
    * Line 521: type “it may expensive in practice”

**Questions:**

* Regarding the objective (section 3.1) and specifically the last part about maximizing the performance on the accept tasks – would it make sense or be beneficial to aim for not changing the original task performance instead of maximizing task performance?

* I could not understand the note in lines 209-210 means. Could you please explain?

---

### Official Review · Reviewer_jM2U · 2024-10-26

**Soundness:** 2
**Presentation:** 4
**Contribution:** 3
**Rating:** 3
**Confidence:** 4

**Summary:**

This paper investigates methods for scoping language models to specific tasks, aiming to make them accept queries from tasks from an 'accept' set, while rejecting queries of tasks from an 'reject' set. The authors evaluate various techniques including system prompting, supervised fine-tuning (SFT), Direct Preference Optimization (DPO), and Circuit Breakers (CB).

They conduct extensive experiments examining robustness to adversarial prompts, data diversity requirements, and multi-task acceptance capabilities. Their results suggest that CB-based methods, especially when combined with SFT, provide robust task scoping while maintaining performance on accepted tasks. The work provides valuable empirical insights into the relative strengths of different scoping approaches, though it has limitations in its problem definition and evaluation methodology.

**Strengths:**

- **Comprehensive Comparative Empirical Evaluation**: The paper provides a thorough comparison of different scoping techniques in a unified setting, including recent methods like DPO and Circuit Breakers. This type of broad evaluation of different is valuable for the field, even without introducing new techniques. As new techniques get introduced so frequently, one often lack unbiased comparisons of the new methods in similar setting.
- **Important Problem Space**: The scoping of LLM tasks is an important direction of research. If solved, it could significantly increase both the usefulness and predictability of LLMs in deployed applications.
- **Clear Research Questions**: The experimental section is generally well-structured around specific, well-motivated research questions, data diversity requirements, and multi-task acceptance. The experimental section are clearly exposed. I particularly appreciated the "Takeaways" paragraph that enabled an easier digestion of the experimental results.
- **Thorough Documentation**: The methodology and experimental setup are extensively documented. The authors consistently include their choice of hyperparameters. The paper includes detailed ablation studies and additional analyses covering aspects like data quantity effects, LoRA rank impact, and representation analysis.

**Weaknesses:**

**Limited Scope of Interventions**: The paper focuses solely on model weight updates (computing Δ), ignoring important external intervention techniques like input-output safeguards or LLM judges that are commonly used for task scoping in practice. See for instance the references:
"NeMo Guardrails: A Toolkit for Controllable and Safe LLM Applications with Programmable Rails", Rebedea et al.
"Llama Guard: LLM-based Input-Output Safeguard for Human-AI Conversations", Inan et al.

**Unrealistic Dataset Selection**: The "accept" tasks (Sentiment Analysis, Summarization, Program Execution) are narrow NLP tasks that don't reflect the diverse, flexible use cases of LLMs in real applications. More realistic datasets like Alpaca are only used as out-of-distribution tests.


**Evaluation Methodology Issues**:
- ROUGE-L score is used inappropriately for creative tasks like story completion where multiple valid outputs exist. In practice as the tasks Sentiment Analysis, Summarization, Program Execution are the main used in the "accept set", this doesn't radically change the main experimental results. However, the figure 4 includes "Story completion", and "Dialogue generation" in the "accept" set. The choice of ROUGE-L lacks motivation, and recognition of its limit for creative tasks.
- The rejection detection system lacks validation on held-out data and uses different detection methods for different techniques (especially CB), potentially biasing results.
- In Figure 2, for the robustness evaluation under adversarial attacks, the task performance is measured _under attack_ (this is why for instance, the task performance is always ~ 0 for the base64 attacks, because the model cannot meaningfully process request encoded in base64). ID reject and OOD Reject should used the adversarial attack, but I don't understand why use the attack to measure the accept task performance and refusal rate.
- The system prompt (Sys.) method only includes a high-level description of the tasks, and doesn't include few shot example of the 'accept' or 'reject' task. This is likely to greatly underestimate the effectiveness of prompting. Even if I expect prompting alone to be less effective than CB and DPO, I would have liked to see more effort put in the prompting baseline.

**Unclear Problem Definition**:

The expected behavior for out-of-distribution tasks is not explicitly justified, and the assumption that all non-accept tasks should be rejected seems artificial and potentially limiting for real applications. In particular, in the case where only one task is included in the "accept", and the "reject" task, it is unclear to my why we should expect the "right" generalisation to be "Accept only the 'accept' task, and reject every other task", and not for instance the symmetric case "Reject only the 'reject' task, and accept every other tasks".

**Questions:**

1. Why is adversarial robustness prioritized for task scoping? Unlike safety applications, is there a compelling use case where preventing a model from performing an unintended but harmless task is critical?
2. For the adversarial evaluation, what is the threat model used? Why measuring the task performance _under attack_?
4. Could you clarify why the probe score is consistently 0 in Figure 2? This seems unusual and warrants explanation.
5. Have you considered evaluating external intervention techniques (like input-output safeguards) as baselines? These are common in practice and their exclusion seems to limit the paper's practical relevance.
6. For creative tasks, have you considered using LLM judges for evaluation instead of ROUGE-L? This could provide more meaningful metrics for tasks with multiple valid outputs.
7. Could you provide validation results for the rejection detection system, particularly on held-out data?
8. How would the system prompt baseline perform if enhanced with few-shot examples of both accept and reject tasks?


Minor typos:

* On line 233. " cannot answer that." -> "I cannot answer that."
* Typo line 284: "respond 11I cannot 285 answer that.”" -> "respond ”I cannot 285 answer that.”"

---

### Official Review · Reviewer_PaVP · 2024-11-02

**Soundness:** 1
**Presentation:** 3
**Contribution:** 2
**Rating:** 3
**Confidence:** 4

**Summary:**

The paper proposes a threat model about language models: they might fail in reducing their scope, ending in generating responses out of their professional scope, or even generating private-sensitive content. The authors propose a new testbed for this threat model, and then analyze and compare current alignment approaches on this testbed.

**Strengths:**

1. **Introduction of a New Threat Model**: The paper proposes a new threat model for scoping language models, which helps reduce potential risks associated with model misuse.
2. **Comprehensive Comparison of Existing Methods**: The authors conduct a comprehensive comparison of current scope-reducing methods across a wide range of tasks, providing insights into each method’s feasibility and limitations.
3. **Intepretability-based Methods Involved**: The authors evaluated interpretability-based methods' performance on reducing the scope of LLMs, which are lightweight and potentially scalable.

**Weaknesses:**

1. **Limited Data Transparency**: I believe that the greatest contribution of this paper is to propose this scope-reducing task and make a solid testbed for evaluating scoping behaviors. However, the testbed and the data they used are not open-sourced, and I don’t see a commitment from the authors to make them available. This will largely decrease the contribution and the soundness of this paper.
2. **Limited Coverage of Overlapping Scopes**: The paper does not address cases where different scopes overlap, which can be common in real-world applications (e.g., a user query about storytelling could potentially involve the leak of privacy, like the famous grandma exploit strategy). Ignoring overlapping scopes limits the study’s relevance, as managing such intersections is crucial for robust scope control in practical deployments, instead of the narrow ones (which only involve binary judgment) provided in the paper.
3. **Lack of Fine-Grained Analysis on Failure Cases**: The paper does not provide an in-depth examination of cases where current methods fail, especially for adversarial prompts. A detailed analysis of failure modes would have been valuable for understanding limitations and improving future scoping methods.
4. **Lack of Ablation across Model Sizes**: The paper does not explore how the effectiveness of scoping methods, particularly interpretability-related methods, and prompt-engineering methods, might vary with different model sizes. Instead, the paper performs its experiments on models with similar size. Since smaller and larger models may respond differently to scoping interventions, an ablation across model sizes would provide valuable insights into the scalability and adaptability of these techniques for various deployment scenarios.

**Questions:**

1. Reject sampling (i.e. train a reward model that prefers only detailed answers on the predefined scope and prefers rejection answers on other scopes, then use it to score the base model output and directly replace low-score outputs with rejection answers or select a BoN) seems like a feasible and straightforward approach for reducing the scope of language models. Why did the authors choose not to explore or implement this method?
2. Does the threat model proposed by the authors have a solid basis, or is it primarily derived from the specific chat logs provided? It would be helpful to understand if there are solid evaluations supporting that failing to reduce LLMs' scope will cause significant external risks beyond the empirical examples shown.

---

### Official Review · Reviewer_bfAi · 2024-11-03

**Soundness:** 3
**Presentation:** 2
**Contribution:** 3
**Rating:** 5
**Confidence:** 4

**Summary:**

The authors present an exciting topic of LLM "scoping", where the goal is to limit/control the LLM w.r.t. a set of allowed ("Accepted") topics for answer generation. This is an important line of research, as many consumer-facing LLMs that should only cover a limited set of particular topics (scope) can be exploited in numerous ways if no guardrails/scoping exists. The paper systematically explores the techniques used for scoping/guardrailing while simultaneously probing those techniques for robustness towards various adversarial prompts. The authors try to show through a set of experiments that, among many techniques using Supervised Fine Tuning (SFT) with Circuit Breakers (CB) on a set of training samples that include topics which should be Accepted or Rejected, yields the most optimal results in terms of the decay of the original task performance (Accepted performance), not rejecting the accepted topics, in-domain Rejected (ID Reject) Topic detection and out-of-domain Rejected detection (OOD rejection). The authors further explore the techniques w.r.t. rejection set diversity and what happens if multiple accept topics are present.

**Strengths:**

The paper presents a novel research line for scoping LLMs into custom topics that are outside of traditional toxicity, harmful text/query rejection and more general and diverse in terms of topic set composition.

The proposed techniques have been meticulously tested w.r.t. robustness against adversarial prompts/instructions and performance in terms of maintaining the strength of the model on the original tasks while adding the capability to filter rejected topics and allow accepted topics (scoping) within LLM generation. The outcomes have exposed the inadequacy of methods such as system prompting and DPO and shown that the novel proposed method that uses SFT with Circuit Breakers is the most potent in the evaluation.

A further set of systematic studies is provided to analyse what happens when the diversity of the rejection topics increases and what the behaviour is in the case of multiple accepted topics.

**Weaknesses:**

1) While the study spans many experimentations, only one relatively small LLM is covered (**Mistral-7B-Instruct-v0.2**). This makes assessing the generality of the overall claims relatively non-straightforward or impossible. While I am aware that running the suite of all of these experimentations is relatively computationally costly for various models at scale, having only a single model (I also saw Granite-7B used, but the results seem different for it, confirming this point) does not allow me to conclude that the presented analysis is generalisable across strictly. What happens as the model scale increases or decreases (2B,3B,13B,30B,100B, 100B+) ? Do the claims stay true? Is CB and SFT+CB as efficient with more or less parameters? Do other methods gain anything at scale?

2) The authors present a method for verifying if the output is rejected by using an early keyword lookup or sequential repetitions ("circuit broken") of the same token. While 30 samples per accept, reject, and OOD reject were chosen for tuning the threshold on these heuristics, it is not obvious how robust this combined method is given the overall diversity within the training/validation set of tasks. The amount of samples does not seem representative for such a diverse dataset. Has the method been extensively tested within SNI?

3) It is very hard to parse all of the figures within the paper due to the visibility, scale and colour schemes used. Also, as the x-y scaling of the sub-figures within the figures is different, it is very hard to assess the changes and differences in overall methods performance (i.e. accept maintained performance vs rejected in-domain and OOD changes/ratio).

4) While SFT + CB seems to be a fairly consistent (with some exceptions) leader in performance w.r.t. all of the objectives (accept performance and not-filtering vs rejected in-domain and OOD filtering) along with being robust to adversarial prompts, the much simpler method of probing is also fairly consistent when rejecting/filtering (in most of the setups) in-domain and OOD rejected samples. The difficulty that it can also over-filter the accepted set seems to be relatively overcome-able by maybe training a larger MLP (or fine-tuning a pre-trained model) or using additional heuristics along with the Probing.
Also, probing seems much better than any method in "multiple accepted topics" evaluation, showing that achieving good scoping can be as simple as training a probing model. With this we might assume that Probing might be a much stronger benchmark than presented. Is there a more extensive comparison of probing techniques with SFT+CB?

Minor Corrections:

line 43: "give give" -> "give"

line 299: "to to" -> "to"

line 457: I assume the text should say "when there are multiple tasks in the *accepted* set"

**Questions:**

**Question set 1**: For context, see point 1 in weaknesses:

*What happens as the model scale increases or decreases (2B,3B,13B,30B,100B, 100B+)?*

*Do the claims stay true?*

*Is CB and SFT+CB as efficient more or less parameters?*

*Do other methods gain or loose anything at scale?*

**Question set 2**: For context, see point 2 in weaknesses:

*Has the rejection verification method been extensively tested within SNI?*

**Question set 3**: For context, see point 4 in weaknesses:

 *Is there a more extensive comparison of probing techniques with SFT+CB?*

---

### Note · Authors · 2024-12-02

I have read and agree with the venue's withdrawal policy on behalf of myself and my co-authors.